# OpenReview forum: "Approximating Human Preferences Using a Multi-Judge Learned System"
_NeurIPS.cc/2025/Workshop/Reliable_ML — NeurIPS 2025 - Reliable ML Workshop_

### Official Review · Reviewer_rT2F · 2025-09-17
**Comment**

**Rating:** 8
**Confidence:** 5

**Review:**

I quite enjoyed reading this paper. The work is well-written, the figures are nice. The framework of using specialized judges and a learned aggregator is interesting.

Suggestions/Weaknesses:
I thought your robustness studies were a great start, covering biased training data and biased judges separately. However, I think a missing robustness question is what happens when a biased/imperfect user interacts with a set of (potentially) biased/imperfect judges.

---

### Official Review · Reviewer_Nzgp · 2025-09-23
**This paper proposes a framework for approximating human preferences by aggregating outputs from multiple rubric-conditioned LLM judges.**

**Rating:** 8
**Confidence:** 4

**Review:**

Summary: To simulate diverse human evaluators, the authors introduce persona-based judges that represent different perspectives, which are then used to generate synthetic preference labels. Two aggregation architectures are studied: a Generalized Additive Model (GAM), which provides interpretability, and a Multi-Layer Perceptron (MLP), which yields slightly stronger predictive accuracy. Across experiments on the UltraFeedback dataset, the learned aggregators improve over heuristics.

Strengths

- Instead of assuming a single “true” human preference, the paper uses a set of 14 personas to capture heterogeneity. This is such a fresh perspective on preference labels, it is also highly impactful as evident in the experimental results.

- Balanced interpretability vs. performance – The GAM offers clear insights into judge contributions (e.g., Truthfulness, Instruction Following, Clarity, Conciseness, and Logical Consistency emerge as top features), while the MLP achieves the best numerical performance. This dual approach ensures both practical utility and analytical depth.

- The experiments go beyond accuracy. The authors have designed and performed experiments to test robustness to random noises, sensitivity to rubrics, and persona variations. I think they have covered most of the dimensions for a deep empirical analysis.

Weaknesses

- Reliance on synthetic ground truth – While personas are clever, the system is ultimately trained against LLM-simulated preference labels, not real human annotations. The authors acknowledge this as a limitation, noting that “we do not yet calibrate to a held-out human-labeled set or report inter-annotator agreement”

Question to think about:

- How might incorporating a small human-annotated set improve the system’s ability to generalize beyond synthetic personas? For example, would the wide 25-point gaps in $R^2$ personas (0.442–0.693) shrink when grounded with real human ratings? or would this gap increase further?